# Trends and influence factors in the prevalence, awareness, treatment, and control of hypertension among US adults from 1999 to 2018

Zhixing Fan[1,2,3], Chaojun Yang[1,2]*, Jing Zhang[1,2], Yifan Huang[1,2], Ying Yang[1,2], Ping Zeng[1,2], Wanyin Cai[1,2], Zujin Xiang[1,2], Jingyi Wu[1,2], Jian Yang[1,2]*

1 Department of Cardiology, the First College of Clinical Medical Sciences, China Three Gorges University, Yichang, China, 2 Institute of Cardiovascular Diseases, China Three Gorges University, Yichang, China, 3 Department of Medical Record Management, the First College of Clinical Medical Sciences, China Three Gorges University, Yichang, China

* yangjian@ctgu.edu.cn (JY); yangchaojun@ctgu.edu.cn (CY)

## Abstract

### Objective

We aimed to describe the trends and influence factors in the prevalence, awareness, treatment, and control of hypertension among US Adults from 1999 to 2018.

### Methods

We utilized data from the National Health and Nutrition Examination Survey (NHANES) spanning ten survey cycles (n = 53,496). Prevalence, awareness, treatment, and control of hypertension were calculated using survey weights. Joinpoint regression and survey-weighted generalized linear models were used to analyze trends and influence factors, respectively.

### Results

The estimated prevalence of hypertension increased significantly from 33.53% to 40.58% (AAPC = 0.896, $P$ = 0.002) during 1999–2018 with dropping rate of newly diagnosed hypertension from 8.62% to 4.82% before 2014 (APC = -4.075, $P$ = 0.001), and then rose to 7.51% in 2018 (APC = 12.302, $P$ = 0.126). Despite modest improvements or stability in the awareness, treatment, and control since 1999, the latter two remained inadequate in 2018 at 59.52% and 51.71%. There was an uptrend in the use of angiotensin-converting enzyme inhibitors (from 24.02% to 45.71%) and angiotensin receptor blockers (from 20.22% to 38.38%), and downtrend in β-blocker (from 12.71% to 4.21%). Men were at higher risk of incidence, un-awareness, un-treatment, and un-control for hypertension. Lower income and education were associated with susceptibility to hypertension, while being married was favorable for treatment and control. Optimal health reduced the incidence of hypertension, and increased the awareness and treatment.

**Data Availability Statement:** All data are available from NHANES (https://www.cdc.gov/nchs/nhanes/index.htm)

**Funding:** This research was supported by the National Natural Science Foundation of China (No. 82371597, 81800258, 82271618), supported by the Natural Science Foundation of Hubei Province (2023AFB609) and supported by Health Commission of Hubei Province (WJ2023M150).

**Competing interests:** The authors have declared that no competing interests exist

**Abbreviations:** CVD, cardiovascular diseases; NHANES, National Health and Nutrition Examination Survey; US, United States; LS7, Life's Simple 7; MEC, mobile examination center; ACEI, angiotensin-converting enzyme inhibitors; ARB, angiotensin receptor blockers; CCB, calcium channel blocker; PIR, income-to-poverty ratio; BMI, body mass index; HEI, healthy eating index; CI, confidence interval; AAPC, average annual percent change.

## Conclusion

Although the rate of newly diagnosed hypertension has declined slightly since 2010 in the US, the prevalence of hypertension is increasing, and treatment and control rates remain inadequate. To manage hypertension effectively, we need to focus on screening and prevention for high-risk populations, while advocating for optimal health to improve the burden of hypertension.

## Introduction

Hypertension is an important risk factor for cardiovascular diseases (CVD), and approximately half of the US adult population meet diagnostic criteria [1]. One study of over 23,000 participants found that half of deaths from coronary heart disease and stroke were among individuals with hypertension [2]. As substantial percentage of patients with un-controlled blood pressure and its strong association with increased CVD risk, especially stroke and heart failure [3, 4], accurately knowing the prevalence, awareness, treatment, and control of hypertension is an important public health issue.

The National Health and Nutrition Examination Survey (NHANES) is a large health and nutritional survey of the civilian noninstitutionalized population of the United States (US) and is extremely useful for monitoring trends in the health status of the population [5]. From NHANES 1999–2000 to 2009–2010, the prevalence of hypertension was stable (from 29.5% to 29.5%), and the rates of awareness and control were improved (from 63.8% to 74.0%; from 27.5% to 46.5%) [6]. From 1999–2000 to 2013–2014, there was a rise in hypertension awareness, treatment, and control [7]. Using the 2015–2016 NHANES survey data, the prevalence of hypertension in the US according to the updated guidelines was 45.4%, corresponding to an estimated 108 million individuals [8]. In NHANES 2017–2018, the prevalence of hypertension was estimated to be 49.64%, and the overall rate of well-controlled hypertension was only 39.64% [9]. Muntner et al [10] found that the prevalence of controlled blood pressure increased from 1999–2000 to 2007–2008, did not significantly change from 2007–2008 through 2013–2014, and then decreased after that. As a result, the prevalence of hypertension is increasing and blood pressure control is inadequate over the years, reflecting an unoptimistic status of hypertension.

The influence factors of hypertension and its control are very important for the reducing the burden of hypertension [11]. The prevalence, awareness, treatment, and control of hypertension were found to differ across racial groups [12]. It is well known that age has a positive correlation with hypertension, while the influence of gender on it is not uniform [13, 14]. Socioeconomic status also plays an important role in hypertension prevalence and its control [15]. The Life's Simple 7 (LS7) metric incorporates health behaviors (body mass index, diet, smoking, physical activity) and health factors (blood pressure, cholesterol, glucose) to estimate an individual's level of cardiovascular health [16]. Plante et al [17]. found that each 1-point improvement in LS7 score was associated with a 6% lower risk of incident hypertension. Thus, race, age, gender, socioeconomic status and LS7 are the crucial factors for the management of hypertension prevention and control.

Using data from 10 National Health and Nutrition Examination Surveys (NHANES), we aimed to update the national trend in the prevalence of hypertension (contained newly and previous diagnosed), assess the tendency of awareness, treatment, and control of hypertension, and further explore their influence factors.

## Material and methods

### Data collection

The NHANES is a nationally representative survey to monitor the health of the US population conducted by the Centers for Disease Control and Prevention, with a complex and multistage sampling design. The NHANES were conducted every 2 years. Participants who were recruited from the US non-institutionalized and civilian population, undergoing 4 stages of selection, including counties, segments, households, and individuals. Data collection was performed through in-home interviews and study visits to a mobile examination center (MEC). The NHANES study protocol was approved by the NCHS Research Ethics Review Board, and written informed consent was obtained from the participants [18]. The additional ethical review was no longer required for the present study due to the usage of publicly available data without identifiable personal information.

Our study included NHANES participants from 1999 to 2018 who who were over 20 years old and were either hypertensive or non-hypertensive (n = 55,043). We excluded pregnant individuals and those with no information on weight variable (n = 1,547). The flow chart of this study was shown in S1 Fig in S1 File.

### Definition of hypertension and its awareness, treatment, and control

Blood pressure determinations were taken at the MEC by a trained physician and measured in the right arm unless specific conditions prohibited the use of the right arm. Three consecutive blood pressures readings were obtained after 5 minutes of quiet rest in a seated position. A fourth attempt may be made if a blood pressure measurement was incomplete or interrupted. The average blood pressure was calculated by the NHANES analytic notes [19], with the following rules: 1) the diastolic reading with zero was not used to calculate the diastolic average; 2) if all diastolic readings were zero, then the average would be zero; 3) if only one blood pressure reading was obtained, that reading was the average; 4) if there was more than one blood pressure reading, the first reading was always excluded from the average. The diagnostic criteria of hypertension [20] was average blood pressure ≥ 140/90 mmHg as well as self-reporting of "ever told you had high blood pressure" or "taking prescription for hypertension". Previous diagnosed hypertension was defined as self-reporting of "ever told you had high blood pressure" or "taking prescription for hypertension" regardless of the current average blood pressure. Newly diagnosed hypertension was defined as average blood pressure ≥ 140/90 mmHg without self-report of hypertension history and drug use.

Awareness of hypertension was defined as a positive reply to the history of hypertension [7]. Treatment of hypertension was defined as an affirmative response to the question of "Are you now taking prescribed medicine for high blood pressure?" [7]. Controlled hypertension was defined as systolic blood pressure < 140 mm Hg and diastolic blood pressure < 90 mm Hg in participants with hypertension [7]. The classification of antihypertensive drugs was obtained during the household interview. If participants answered affirmatively to the question "if you had taken any prescription medications in the past 30 days", they should show the medication containers of all the products used or report the name of the medication to the interviewer. When the interviewer entered the medication name into the computer, the name was automatically identified as either an exact match or a similar text matches. Using drug-classification codes, we determined the classification of antihypertensive medication, including angiotensin-converting enzyme inhibitors (ACEI), angiotensin receptor blockers (ARB), calcium channel blocker (CCB), β-blocker and diuretic (S1 Table in S1 File).

## Social demography and LS7

The sociodemographic factors consisted of age, sex (male and female) and race (non-Hispanic white, non-Hispanic black, Mexican American, non-Hispanic Asian and other race), marital status (yes and no), education level (less than high school, high school graduate, some college, and college graduate or above), insurance status (uninsured and any insurance), income-to-poverty ratio (PIR). PIR was divided into three categories: PIR <1, low income; PIR 1–1.8, middle income; PIR≥1.8, high income.

LS7 scores were calculated by 7 components of individual health behaviors or factors, including average blood pressure, total cholesterol, HbA1c/diabetes, smoking status, body mass index (BMI), physical activity, healthy eating index (HEI). Each individual component was scored as 0 points (poor), 1 point (intermediate), or 2 points (ideal) (S2 Table in S1 File). A total score was calculated as the sum of the points from all LS7 components, ranging from 0 to 14. Participants were categorized into three LS7 health categories [17]: scores ≤4 points indicated inadequate health, 5 to 9 indicated average health, and 10 to 14 indicated optimal health.

## Statistical analysis

NHANES data was extracted and preprocessed by "nhanesR package". NHANES provided weights to ensure a representative and unbiased estimation of the total civilian non-institutionalized US population. The prevalence was weighted by 2-year weights from 1999 to 2018. While we combined 20-year weights to calculate the total prevalence and analyze the influence factors.

The estimation of weighted prevalence and proportion were both conducted by "survey package" that was specially handled complex sample design surveys, summarized as mean and 95% confidence interval (CI). Joinpoint regressions was used to determine trends in log-transformed prevalence, awareness, treatment, and control of hypertension, allowing 1 joinpoint. The Monte Carlo method was used for selecting the best-fitting model and identifying point of change in trends (joinpoint) [21]. Annual percent change (APC) was used to evaluate the internal trend of each independent interval before and after inflection. If there was no turning point, average annual percent change (AAPC) and its 95% CI were calculated to express the trend in the prevalence, awareness, treatment, and control of hypertension.

$$AAPC \ = \ \left(e^{\Sigma w_i \beta_i / \Sigma w i} - 1\right) \ \times \ 100$$

Where $w_i$ is the interval span width of each piece-wise function (i.e., the number of years included in the interval), $\beta_i$ is the regression coefficient. When the AAPC and its 95% CI were both greater than 0, the rate indicated an uptrend; on the contrary, when the AAPC and its 95% CI were both less than 0, the rate indicated a downtrend; any AAPC with a 95% CI overlapping with zero was considered stable trend.

Survey-weighted generalized linear models were performed to evaluate odds ratio (OR) and 95% CI by "survey package" (family = quasibinomial), estimating the factors associated with the prevalence of hypertension, and the factors for the awareness, treatment, and control of hypertension.

Statistical analysis was performed using R version 4.1.1 software and Joinpoint regressions Program 4.9.0.0. Statistical significance was < 0.05.

## Result

### Trends in the prevalence of hypertension

To analyze the trend in the prevalence of hypertension, 53,496 participants were included, representing an estimated 214,087,600 US adults aged 20 or older. A total of 22,947 patients satisfied the diagnostic criteria for hypertension, with a prevalence of 37.56%. The estimated prevalence of hypertension increased significantly from 33.53% to 40.58% from 1999 to 2018, with AAPC of 0.896 ($P$ = 0.002 for trend) (Fig 1 and Table 1). A significant increase in the estimated prevalence of hypertension was observed in the following population over the study period (Table 1): male (from 32.45% to 42.55%; $P$<0.001 for trend), non-Hispanic white (from 33.56% to 41.87%; $P$ = 0.008 for trend), non-Hispanic black (from 40.21% to 49.37%; $P$<0.001 for trend), Mexican American (from 25.01% to 27.49%; $P$ = 0.015 for trend), other Hispanic (from 29.39% to 34.94%; $P$ = 0.029 for trend), married (from 36.10% to 42.85%; $P$ = 0.044 for trend) or not (from 31.40% to 37.99%; $P$ = 0.003 for trend), high school graduate (from 37.60% to 43.44%; $P$ = 0.008 for trend), some college (from 30.59% to 42.83%; $P$<0.001 for trend), college graduate or above (from 25.87% to 34.24%; $P$ = 0.027 for trend), low income (from 34.56% to 41.33%; $P$ = 0.005 for trend), high income (from 32.00% to 40.42%; $P$ = 0.018 for trend), insured adults (from 36.66% to 42.42%; $P$ = 0.009 for trend) or not (from 20.19% to 29.27%; $P$ = 0.003 for trend), inadequate health (from 68.70% to 80.17%; $P$<0.001 for trend), and average health (from 37.87% to 49.23%; $P$ = 0.005 for trend). The inflection point, the annual rate of change before or after inflection, and AAPC in the prevalence of hypertension were shown in S3 Table in S1 File.

The prevalence of previously diagnosed hypertension and newly diagnosed hypertension were 31.04% and 6.48%, respectively. The previous diagnosed hypertension (from 24.84% to 33.03%) was increasing significantly with 1.544% relative increase per 2-cycle, from 1999–2000 to 2017–2018 (S4 and S5 Tables in S1 File and Fig 1); while, the incidence of newly diagnosed hypertension was significantly dropped from 8.62% to 4.82% during 1999–2014 (APC = -4.075, $P$ = 0.001), and then rose to 7.51% in 2018 (APC = 12.302, $P$ = 0.126) (S6 and S7 Tables in S1 File and Fig 1). In addition to the participants of college graduate or above and optimal health, the other subpopulations were all significantly raised in the trend of previous diagnosed hypertension prevalence from 1999 to 2018 (all $P$<0.05 for trend).

### Analysis of influence factors for hypertension

Establishing survey-weighted generalized linear models, the association between prevalence of hypertension and sociodemographic and LS7 was estimated (Table 2). The prevalence of hypertension was higher associated with 40–59 years old (OR = 2.924; 95%CI: 2.697, 3.170), 60–85 years old (OR = 9.147; 95%CI: 8.380, 9.983), male (OR = 1.090; 95%CI: 1.030, 1.154), non-Hispanic black (OR = 1.521; 95%CI: 1.420, 1.629), and insured (OR = 1.455; 95%CI: 1.342, 1.577). While, Mexican American (OR = 0.782; 95%CI: 0.708, 0.863), other Hispanic (OR = 0.882; 95%CI: 0.782, 0.995), college graduate or above (OR = 0.863; 95%CI: 0.789, 0.944), high income (OR = 0.912; 95%CI: 0.845, 0.984), average health (OR = 0.355; 95%CI: 0.322, 0.391), and optimal health (OR = 0.081; 95%CI: 0.072, 0.092) decreased the risk of hypertension. The factors associated with the prevalence of previously diagnosed hypertension was basically in line with total hypertension. The older and non-Hispanic black were associated with a higher risk of both previously and newly diagnosed hypertension; while, average health and optimal health both showed inverse associations with them.

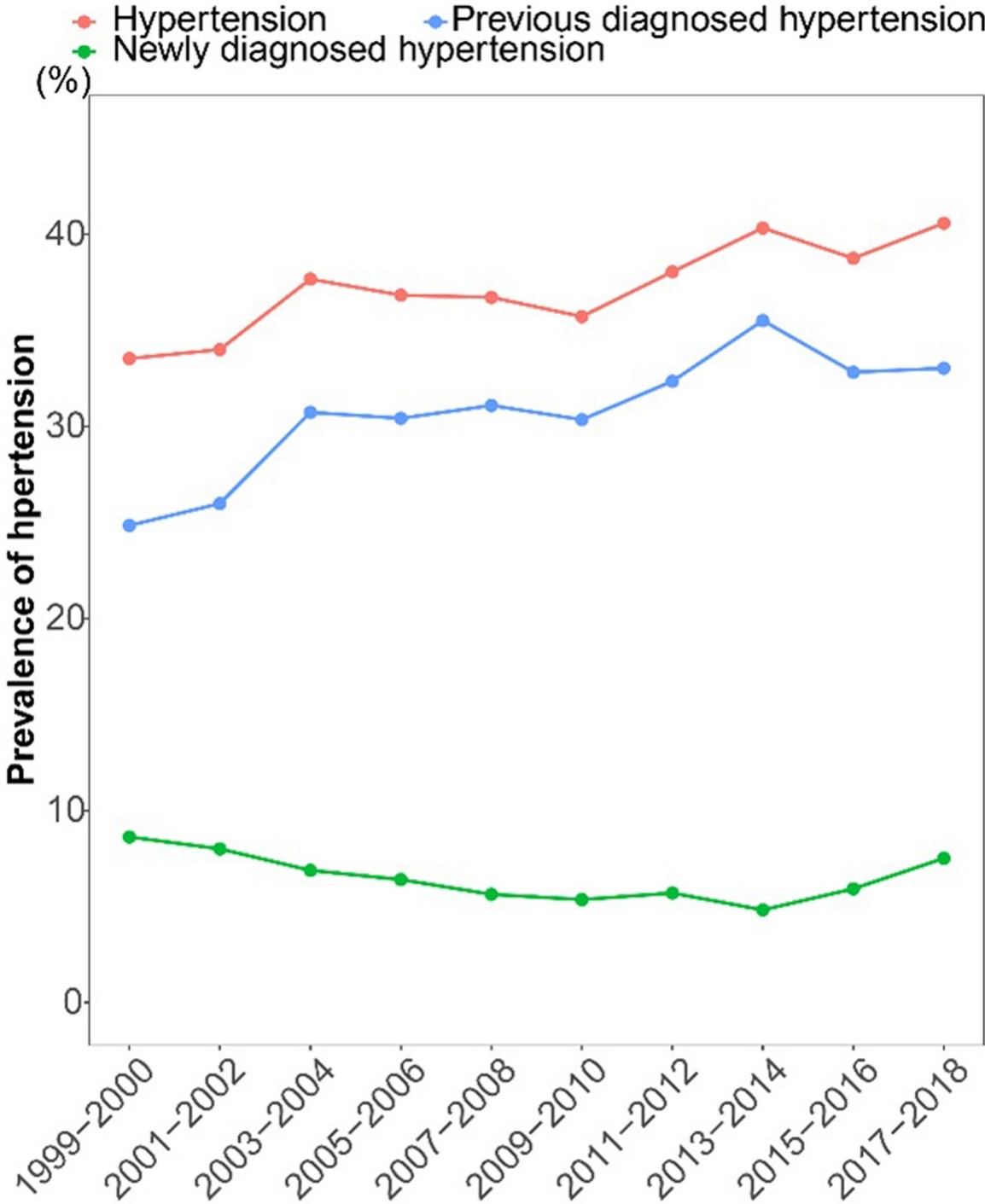

**Fig 1. Trends in the prevalence of hypertension among US adults from 1999 to 2018.**

## Trends in the prevalence of awareness, treatment, and control among hypertension

Among hypertension, the prevalence of awareness, treatment, and control were 81.97%, 60.00%, and 52.46%, respectively. The prevalence of awareness increased from 73.22% to

**Table 1. Prevalence of hypertension among US adults from 1999 to 2018.**

| Characteristics | | Total (n = 53496) | 1999–2000 (n = 4582) | 2001–2002 (n = 5078) | 2003–2004 (n = 4802) | 2005–2006 (n = 4641) | 2007–2008 (n = 5877) | 2009–2010 (n = 6148) | 2011–2012 (n = 5503) | 2013–2014 (n = 5704) | 2015–2016 (n = 5648) | 2017–2018 (n = 5513) | AAPC | P |
|---|---|---|---|---|---|---|---|---|---|---|---|---|---|---|
| Total | | 37.56 (36.01,39.10) | 33.53 (30.41,36.65) | 34.00 (31.41,36.58) | 37.67 (35.38,39.97) | 36.83 (34.41,39.25) | 36.72 (34.65,38.79) | 35.72 (32.84,38.60) | 38.05 (35.37,40.74) | 40.33 (38.25,42.42) | 38.75 (36.10,41.40) | 40.58 (37.34,43.81) | 0.896 (0.42,1.375) | **0.002** |
| Age (years) | 20–49 | 14.94 (14.13,15.76) | 13.21 (9.53,16.89) | 15.20 (13.13,17.26) | 15.62 (12.85,18.40) | 13.84 (10.62,17.06) | 13.85 (12.18,15.53) | 13.31 (11.43,15.18) | 14.41 (12.53,16.30) | 17.06 (14.75,19.36) | 15.14 (12.76,17.53) | 17.28 (14.19,20.38) | 0.728 (-0.52,1.991) | 0.216 |
| | 40–59 | 38.51 (37.29,39.73) | 36.87 (30.90,42.84) | 34.01 (30.46,37.57) | 39.60 (35.60,43.59) | 38.96 (34.75,43.18) | 38.54 (36.01,41.07) | 34.82 (31.19,38.45) | 39.96 (37.12,42.80) | 41.30 (38.26,44.34) | 39.56 (35.96,43.16) | 41.01 (35.63,46.39) | 0.749 (-0.023,1.526) | 0.056 |
| | 60–85 | 68.80 (67.69,69.91) | 67.21 (64.13,70.29) | 69.74 (66.19,73.30) | 70.68 (67.77,73.59) | 69.15 (66.57,71.73) | 69.14 (66.55,71.74) | 69.74 (66.17,73.30) | 67.45 (63.93,70.97) | 69.41 (66.70,72.13) | 67.30 (62.70,71.90) | 68.15 (64.24,72.05) | -0.05 (-0.288,0.188) | 0.639 |
| Sex | Female | 37.61 (36.68,38.55) | 34.57 (31.43,37.70) | 35.91 (33.11,38.71) | 38.26 (35.13,41.39) | 36.97 (33.99,39.95) | 36.92 (34.33,39.51) | 35.25 (32.70,37.80) | 37.31 (34.68,39.94) | 40.55 (37.40,43.71) | 38.03 (34.84,41.22) | 38.70 (35.38,42.02) | 0.503 (-0.011,1.019) | 0.054 |
| | Male | 37.50 (36.48,38.51) | 32.45 (28.35,36.55) | 31.99 (28.72,35.26) | 37.06 (34.35,39.77) | 36.69 (33.89,39.48) | 36.51 (33.86,39.16) | 36.21 (32.42,40.00) | 38.85 (35.39,42.30) | 40.09 (38.06,42.13) | 39.51 (36.76,42.26) | 42.55 (38.48,46.63) | 1.231 (0.738,1.726) | **<0.001** |
| Race | Non-Hispanic white | 38.62 (37.58,39.66) | 33.56 (29.81,37.31) | 34.89 (32.14,37.64) | 39.16 (36.35,41.97) | 37.87 (34.83,40.91) | 38.13 (35.16,41.10) | 36.70 (32.94,40.46) | 39.06 (35.42,42.70) | 42.70 (39.84,45.55) | 39.34 (36.17,42.51) | 41.87 (37.53,46.21) | 0.972 (0.336,1.611) | **0.008** |
| | Non-Hispanic black | 46.13 (44.99,47.27) | 40.21 (37.70,42.72) | 44.31 (39.32,49.30) | 43.65 (40.35,46.95) | 45.68 (42.91,48.45) | 44.48 (40.61,48.35) | 45.15 (41.43,48.87) | 47.96 (44.62,51.30) | 48.85 (45.55,52.16) | 46.82 (43.02,50.61) | 49.37 (45.46,53.28) | 1.003 (0.618,1.39) | **<0.001** |
| | Mexican American | 25.88 (24.34,27.42) | 25.01 (21.71,28.31) | 19.11 (16.79,21.42) | 23.82 (16.60,31.03) | 23.13 (19.45,26.81) | 24.15 (20.57,27.74) | 25.61 (22.04,29.18) | 27.11 (21.19,33.03) | 27.46 (23.07,31.86) | 31.55 (25.51,37.59) | 27.49 (23.38,31.60) | 1.683 (0.415,2.968) | **0.015** |
| | Other Hispanic | 30.09 (28.01,32.16) | 29.39 (21.59,37.18) | 27.24 (21.28,33.20) | 29.22 (13.99,44.45) | 22.65 (14.80,30.51) | 29.38 (24.61,34.16) | 27.08 (23.75,30.41) | 31.18 (24.89,37.48) | 29.23 (25.28,33.18) | 32.18 (27.20,37.17) | 34.94 (29.70,40.18) | 1.31 (0.173,2.46) | **0.029** |
| | Other race | 32.90 (30.83,34.97) | 36.63 (24.23,49.04) | 26.59 (18.30,34.89) | 30.52 (25.01,36.03) | 32.60 (26.40,38.80) | 28.92 (21.61,36.24) | 29.38 (23.65,35.11) | 31.48 (27.98,34.97) | 31.27 (26.51,36.03) | 36.19 (30.00,42.38) | 37.98 (32.12,43.84) | 1.124 (-0.116,2.379) | 0.07 |
| Marital status | No | 36.05 (34.97,37.13) | 31.40 (28.15,34.65) | 34.21 (30.75,37.67) | 34.68 (30.88,38.47) | 35.87 (32.18,39.56) | 34.23 (31.25,37.20) | 32.96 (29.75,36.17) | 36.49 (33.42,39.56) | 38.70 (35.34,42.06) | 39.02 (35.23,42.80) | 37.99 (34.73,41.26) | 0.984 (0.442,1.529) | **0.003** |
| | Yes | 38.87 (37.91,39.82) | 36.10 (32.26,39.94) | 33.86 (30.44,37.27) | 39.92 (37.74,42.11) | 37.45 (34.73,40.18) | 38.65 (36.67,40.64) | 37.83 (34.50,41.16) | 39.46 (36.05,42.87) | 41.64 (39.30,43.97) | 38.53 (35.60,41.45) | 42.85 (39.14,46.57) | 0.685 (0.022,1.352) | **0.044** |
| Education level | Less than high school | 43.65 (42.33,44.96) | 39.42 (36.24,42.60) | 43.34 (38.39,48.30) | 45.06 (39.87,50.26) | 43.35 (39.74,46.96) | 42.70 (38.62,46.79) | 42.29 (38.15,46.43) | 46.86 (43.89,49.82) | 43.66 (39.79,47.53) | 43.70 (39.34,48.06) | 44.52 (38.66,50.37) | 0.578 (-0.039,1.2) | 0.063 |
| | High school graduate | 40.92 (39.63,42.20) | 37.60 (32.81,42.38) | 35.31 (30.70,39.92) | 41.27 (38.44,44.10) | 39.07 (35.06,43.07) | 38.96 (35.83,42.09) | 39.40 (35.80,43.00) | 41.11 (35.69,46.53) | 45.73 (42.71,48.75) | 44.72 (40.83,48.60) | 43.44 (38.80,48.09) | 1.084 (0.368,1.806) | **0.008** |
| | Some college | 37.13 (35.95,38.32) | 30.59 (26.96,34.23) | 31.45 (29.21,33.68) | 35.08 (32.03,38.13) | 34.01 (30.69,37.34) | 35.24 (31.94,38.54) | 36.19 (33.18,39.20) | 37.56 (32.23,42.88) | 41.93 (37.91,45.95) | 40.53 (37.21,43.85) | 42.83 (38.29,47.38) | 1.838 (1.422,2.256) | **<0.001** |
| | College graduate or above | 31.17 (29.74,32.61) | 25.87 (22.16,29.58) | 28.62 (24.04,33.21) | 31.02 (26.60,35.44) | 33.44 (28.29,38.58) | 31.33 (27.77,34.90) | 27.54 (22.30,32.78) | 31.79 (27.58,36.00) | 33.00 (30.22,35.79) | 30.85 (26.46,35.24) | 34.24 (28.67,39.82) | 0.995 (0.147,1.849) | **0.027** |
| PIR | Low income | 36.07 (34.45,37.69) | 34.56 (28.79,40.32) | 34.09 (29.06,39.12) | 33.00 (28.98,37.02) | 33.46 (28.20,38.72) | 34.63 (30.54,38.72) | 33.47 (30.65,36.30) | 33.63 (27.45,39.80) | 38.91 (36.15,41.67) | 40.00 (34.30,45.69) | 41.33 (35.56,47.09) | 1.286 (0.499,2.078) | **0.005** |
| | Middle income | 41.05 (39.77,42.33) | 37.59 (33.28,41.90) | 37.31 (33.24,41.38) | 40.07 (34.68,45.47) | 42.89 (39.20,46.57) | 39.71 (36.58,42.83) | 40.52 (36.04,45.00) | 42.29 (37.54,47.04) | 44.58 (41.30,47.86) | 41.94 (38.61,45.27) | 39.80 (35.79,43.80) | 0.567 (-0.109,1.248) | 0.089 |
| | High income | 36.87 (35.90,37.85) | 32.00 (28.49,35.51) | 33.10 (30.16,36.05) | 37.99 (35.32,40.67) | 35.80 (33.10,38.50) | 36.36 (34.12,38.60) | 35.28 (31.71,38.85) | 38.09 (34.97,41.21) | 39.41 (36.46,42.35) | 37.14 (34.40,39.89) | 40.42 (36.32,44.52) | 0.827 (0.187,1.47) | **0.018** |
| Health insurance | No | 24.77 (23.55,25.98) | 20.19 (16.35,24.02) | 20.71 (16.03,25.38) | 24.84 (21.40,28.28) | 24.92 (20.59,29.25) | 24.71 (22.58,26.84) | 23.09 (19.60,26.58) | 25.73 (22.33,29.13) | 26.20 (23.49,28.91) | 27.16 (22.59,31.72) | 29.27 (23.94,34.60) | 1.45 (0.674,2.232) | **0.003** |
| | Yes | 40.32 (39.39,41.25) | 36.66 (32.91,40.41) | 36.93 (34.25,39.61) | 40.58 (38.09,43.06) | 39.53 (36.98,42.07) | 39.61 (37.21,42.02) | 39.07 (36.23,41.91) | 41.11 (37.82,44.40) | 43.42 (40.89,45.94) | 40.47 (37.27,43.67) | 42.42 (39.01,45.82) | 0.718 (0.235,1.203) | **0.009** |
| Life's simple 7 | Inadequate health | 71.54 (69.92,73.16) | 68.70 (61.14,76.25) | 63.35 (56.47,70.24) | 67.84 (63.95,71.72) | 64.80 (58.75,70.86) | 70.81 (65.03,76.59) | 71.44 (67.20,75.69) | 71.49 (67.30,75.68) | 72.86 (69.00,76.73) | 81.56 (78.52,84.59) | 80.17 (76.02,84.32) | 1.368 (0.805,1.934) | **<0.001** |
| | Average health | 43.20 (42.30,44.10) | 37.87 (35.36,40.39) | 38.60 (35.85,41.36) | 41.21 (38.11,44.31) | 42.12 (39.57,44.67) | 43.61 (41.52,45.70) | 43.75 (40.70,46.80) | 47.03 (44.26,49.79) | 39.84 (37.12,42.57) | 46.46 (43.77,49.15) | 49.23 (45.81,52.65) | 1.168 (0.457,1.883) | **0.005** |
| | Optimal health | 11.22 (10.38,12.06) | 7.13 (4.78,9.49) | 8.65 (6.42,10.88) | 13.11 (10.41,15.80) | 12.16 (9.76,14.55) | 12.62 (10.43,14.81) | 10.91 (8.33,13.49) | 12.70 (10.50,14.89) | 6.41 (4.02,8.81) | 11.58 (9.06,14.11) | 12.26 (8.81,15.70) | 0.877 (-2.097,3.941) | 0.52 |

PIR: income-to-poverty ratio; AAPC: average annual percent change

87.52% through 1999–2014 (APC = 1.1272, *P* = 0.003), and then decreased to 81.16% in 2018 (APC = -2.101, *P* = 0.276) (S8 and S9 Tables in S1 File and Fig 2). The rates of treatment and control were both increased from 51.16% to 67.47% (APC = 3.053, *P* = 0.001) and from 41.20% to 60.05% (APC = 3.690, *P* = 0.001) before 2009, then undulate down to 59.52% (APC = -1.242, *P* = 0.058) and 51.71% (APC = -1.753, *P* = 0.084) in 2018 after that (S10-S13 Tables

**Table 2. Influencing factors of hypertension among US adults from 1999 to 2018.**

| Characteristics | | Diagnosis of hypertension | | Previous diagnosis of hypertension | | Newly diagnosed hypertension | |
|---|---|---|---|---|---|---|---|
| | | OR (95% CI) | *P* | OR (95% CI) | *P* | OR (95% CI) | *P* |
| Age (years) | 20–49 | 1 (reference) | | 1 (reference) | | 1 (reference) | |
| | 40–59 | 2.924(2.697,3.170) | <**0.0001** | 3.143(2.881, 3.428) | <**0.0001** | 2.183(1.893,2.518) | <**0.0001** |
| | 60–85 | 9.147(8.380,9.983) | <**0.0001** | 9.823(8.977,10.749) | <**0.0001** | 6.544(5.579,7.675) | <**0.0001** |
| Sex | Female | 1 (reference) | | 1 (reference) | | 1 (reference) | |
| | Male | 1.090(1.030,1.154) | **0.003** | 1.012(0.951, 1.077) | 0.703 | 1.330(1.192,1.485) | <**0.0001** |
| Race | Non-Hispanic white | 1 (reference) | | 1 (reference) | | 1 (reference) | |
| | Non-Hispanic black | 1.521(1.420,1.629) | <**0.0001** | 1.568(1.460, 1.684) | <**0.0001** | 1.324(1.145,1.532) | <**0.001** |
| | Mexican American | 0.782(0.708,0.863) | <**0.0001** | 0.731(0.658, 0.812) | <**0.0001** | 0.985(0.838,1.157) | 0.850 |
| | Other Hispanic | 0.882(0.782,0.995) | **0.041** | 0.870(0.763, 0.992) | **0.038** | 0.925(0.708,1.207) | 0.562 |
| | Other race | 1.030(0.924,1.149) | 0.592 | 0.999(0.890, 1.120) | 0.980 | 1.136(0.928,1.392) | 0.215 |
| Marital status | No | 1 (reference) | | 1 (reference) | | 1 (reference) | |
| | Yes | 0.976(0.914,1.041) | 0.456 | 0.987(0.921, 1.058) | 0.712 | 0.915(0.813,1.030) | 0.140 |
| Education level | Less than high school | 1 (reference) | | 1 (reference) | | 1 (reference) | |
| | High school graduate | 1.018(0.939,1.104) | 0.662 | 1.020(0.935, 1.113) | 0.648 | 1.053(0.922,1.203) | 0.445 |
| | Some college | 1.023(0.938,1.114) | 0.608 | 1.035(0.944, 1.134) | 0.465 | 0.957(0.824,1.112) | 0.565 |
| | College graduate or above | 0.863(0.789,0.944) | **0.001** | 0.836(0.759, 0.921) | <**0.001** | 0.981(0.825,1.166) | 0.823 |
| Health insurance | No | 1 (reference) | | 1 (reference) | | 1 (reference) | |
| | Yes | 1.455(1.342,1.577) | <**0.0001** | 1.563(1.427, 1.713) | <**0.0001** | 1.101(0.949,1.279) | 0.202 |
| PIR | Low income | 1 (reference) | | 1 (reference) | | 1 (reference) | |
| | Middle income | 1.020(0.932,1.117) | 0.662 | 1.001(0.918, 1.092) | 0.975 | 1.078(0.894,1.300) | 0.427 |
| | High income | 0.912(0.845,0.984) | **0.019** | 0.863(0.795, 0.936) | <**0.001** | 1.109(0.946,1.300) | 0.200 |
| Life's simple 7 | Inadequate health | 1 (reference) | | 1 (reference) | | 1 (reference) | |
| | Average health | 0.355(0.322,0.391) | <**0.0001** | 0.372(0.334, 0.414) | <**0.0001** | 0.322(0.286,0.362) | <**0.0001** |
| | Optimal health | 0.081(0.072,0.092) | <**0.0001** | 0.095(0.083, 0.108) | <**0.0001** | 0.041(0.032,0.053) | <**0.0001** |

PIR: income-to-poverty ratio

in S1 File and Fig 2). The increased trends in the awareness of hypertension were mainly in the subpopulation of 20–49 years old, 60–85 years old, both female and male, non-Hispanic white, married or not, less than high school, some college, all PIR, insured or not, inadequate health and average health (S8 and S9 Tables in S1 File). The trend in the prevalence of treatment among hypertension raised in 20–49 years old, non-Hispanic black, Mexican American, no marital and Some college (S10 and S11 Tables in S1 File). The prevalence of control among hypertension were increased in the population of 60–85 years old, male, non-Hispanic white, married or not, middle income, insured and average health (S12 and S13 Tables in S1 File).

We further estimated the trends in the prevalence of five antihypertensive agent during study interval. Of those who were now taking prescribed medicine for hypertension, only 3,295 participants reported or showed the specific medicine. The use of ACEI, ARB, CCB, β-blocker and diuretic were 35.36%, 33.21%, 16.32%, 39.80%, 8.78%, and 84.97%, respectively. There was an uptrend in the use of ACEI (from 24.02% to 45.71%, AAPC = 3.368, *P* = 0.001) and ARB (from 20.22% to 38.38%, AAPC = 2.746, *P* = 0.004), and a downtrend in the use of β-blocker (from 12.71% to 4.21%, AAPC = -6.463, *P* = 0.009) from 1999 to 2018 (S14 and S15 Tables in S1 File and Fig 3). Before 2005, the use of CCB and diuretics increased from 6.46% to 23.70 (APC = 24.884, *P* = 0.093) and decreased from 93.54% to 81.18% (APC = -3.208,

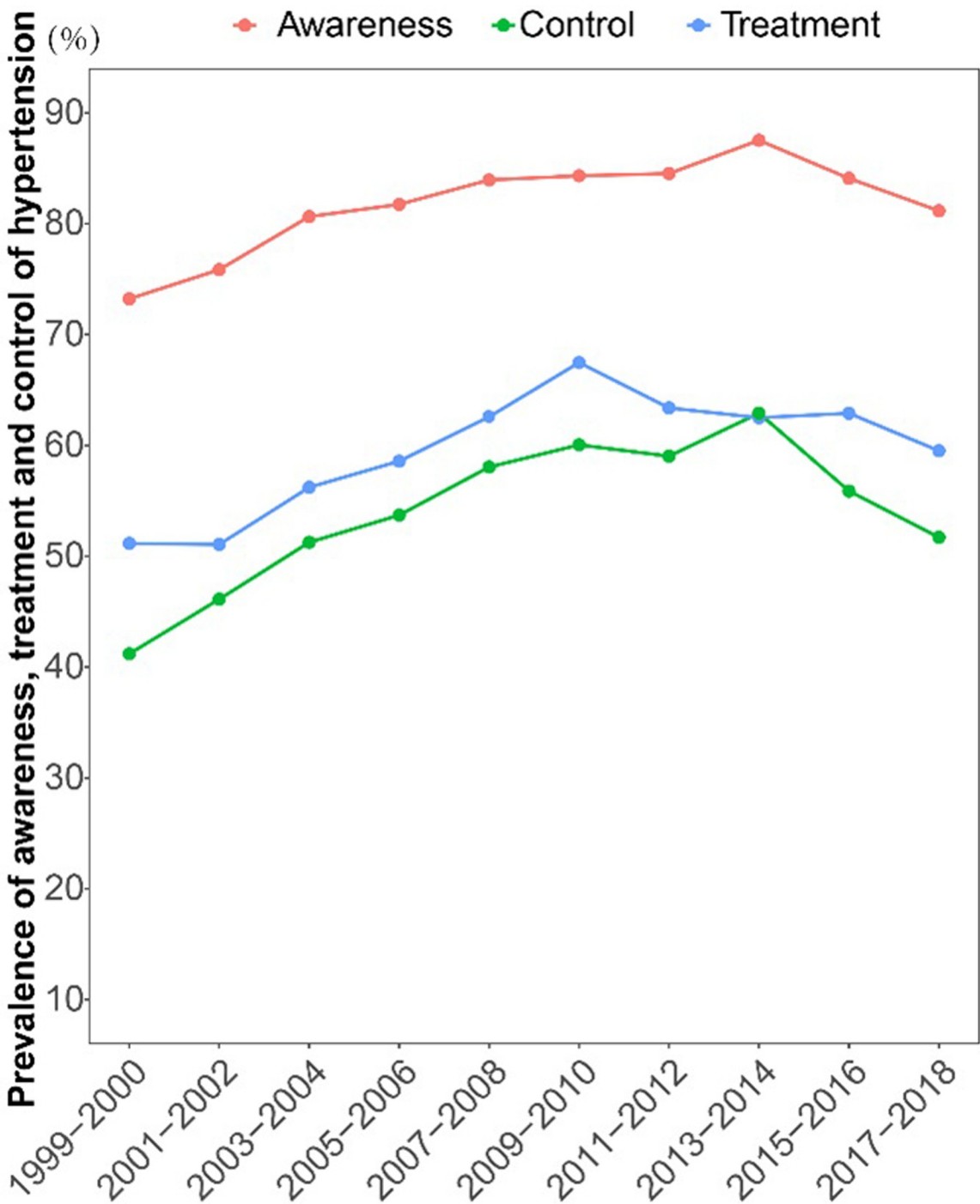

**Fig 2. Trends in the awareness, treatment, and control of hypertension among US adults from 1999 to 2018.**

*P* = 0.081), respectively, and then decreased to 9.05% (APC = -6.665, *P* = 0.021) and increased to 91.89% (APC = 1.231, *P* = 0.022) in 2018 after that.

## Analysis of influence factors for awareness, treatment, and control of hypertension

The influence factors of awareness, treatment, and control of hypertension were displayed in Table 3. The 40–59 years old (OR = 1.388; 95%CI: 1.197,1.608), 60–85 years old (OR = 1.415; 95%CI: 1.216, 1.647), non-Hispanic black (OR = 1.220; 95%CI: 1.063, 1.400), insured (OR = 1.428; 95%CI: 1.213, 1.680), and optimal health (OR = 2.100; 95%CI: 1.623, 2.717) increased the risk of hypertension awareness; while, male (OR = 0.789; 95%CI: 0.704, 0.883), Mexican American (OR = 0.753; 95%CI: 0.653, 0.868) and high income (OR = 0.788; 95%CI: 0.674, 0.921) decreased it. The treatment for hypertension was raised mainly in the population of 40–59 years old (OR = 4.249; 95%CI: 3.692, 4.891), 60–85 years old (OR = 8.661; 95%CI: 7.518, 9.978), non-Hispanic black (OR = 1.537; 95%CI: 1.378, 1.714), married (OR = 1.192; 95%CI: 1.086, 1.309) and insured (OR = 2.313; 95%CI: 2.019, 2.649), and declined with male (OR = 0.767; 95%CI: 0.707, 0.833) and Mexican American (OR = 0.832; 95%CI: 0.725, 0.955). The un-control of blood pressure among hypertension was associated with 40–59 years old (OR = 0.907; 95%CI: 0.802, 1.027)), 60–85 years old (OR = 0.655; 95%CI: 0.579, 0.740), male (OR = 0.891; 95%CI: 0.822, 0.966), non-Hispanic black (OR = 0.810; 95%CI: 0.736, 0.891), Mexican American, (OR = 0.846; 95%CI: 0.738, 0.971); and inversely associated with married (OR = 1.095; 95%CI: 1.008, 1.189), insured (OR = 1.364; 95%CI: 1.195, 1.557), average health (OR = 2.035; 95%CI: 1.834,2.258) and optimal health (OR = 6.669; 95%CI: 5.522,8.055). Subsequently, we explored the influence factors for the control of blood pressure in hypertension with prescription medicine. The control of blood pressure worsened in participants of 60–85 years old (OR = 1.915; 95%CI: 1.495, 2.454), non-Hispanic black (OR = 1.436; 95%CI: 1.299,1.587), and better in insured (OR = 0.769; 95%CI: 0.627, 0.945), average health (OR = 0.479; 95%CI: 0.423, 0.542), and optimal health (OR = 0.124; 95%CI: 0.091, 0.169).

## Discussion

In this study, we analyzed NHANES and found that the rate of newly diagnosed hypertension in the US has slightly declined since 2010, while the prevalence of hypertension is on the rise. Although antihypertension strategy was becoming more standardized with an uptrend in the use of ACEI and ARB, and downtrend in β-blocker, the control and treatment of hypertension are still inadequate. The trends and influences in prevalence, awareness, treatment, and control of hypertension varied with age, gender, race, education, income, and LS7. Therefore, in the management of hypertension, the focus population should be screened out and stratified by age, sex and race for precise prevention and control. Meanwhile, optimal health should be advocated to improve the burden of hypertension.

The estimated prevalence of newly diagnosed hypertension decreased from 8.62% to 7.51% during 1999–2018, while the whole prevalence of hypertension increased significantly from 33.53% to 40.58%, and the treatment and control in hypertension remained at 59.52% and 51.71% in 2018, respectively. It shows that although the hypertension prevention and control measures taken in the past have achieved a certain effect, the burden of hypertension is still relatively heavy. In terms of medication for hypertension, there was an uptrend in the use of ACEI (from 24.02% to 45.71%) and ARB (from 20.22% to 38.38%), and downtrend in the use of β-blocker (from 12.71% to 4.21%), which is consistent with recommendations in the guidelines for the treatment of adult hypertension drugs [22]. While, we also found that the trend in the prevalence of CCB and diuretic remained stable. In fact, a growing body of clinical evidence suggests that diuretics and CCB are essential for effective blood pressure control in older adults [23, 24]. With the increasing degree of population aging, the clinical use of diuretics and CCB needs more attention.

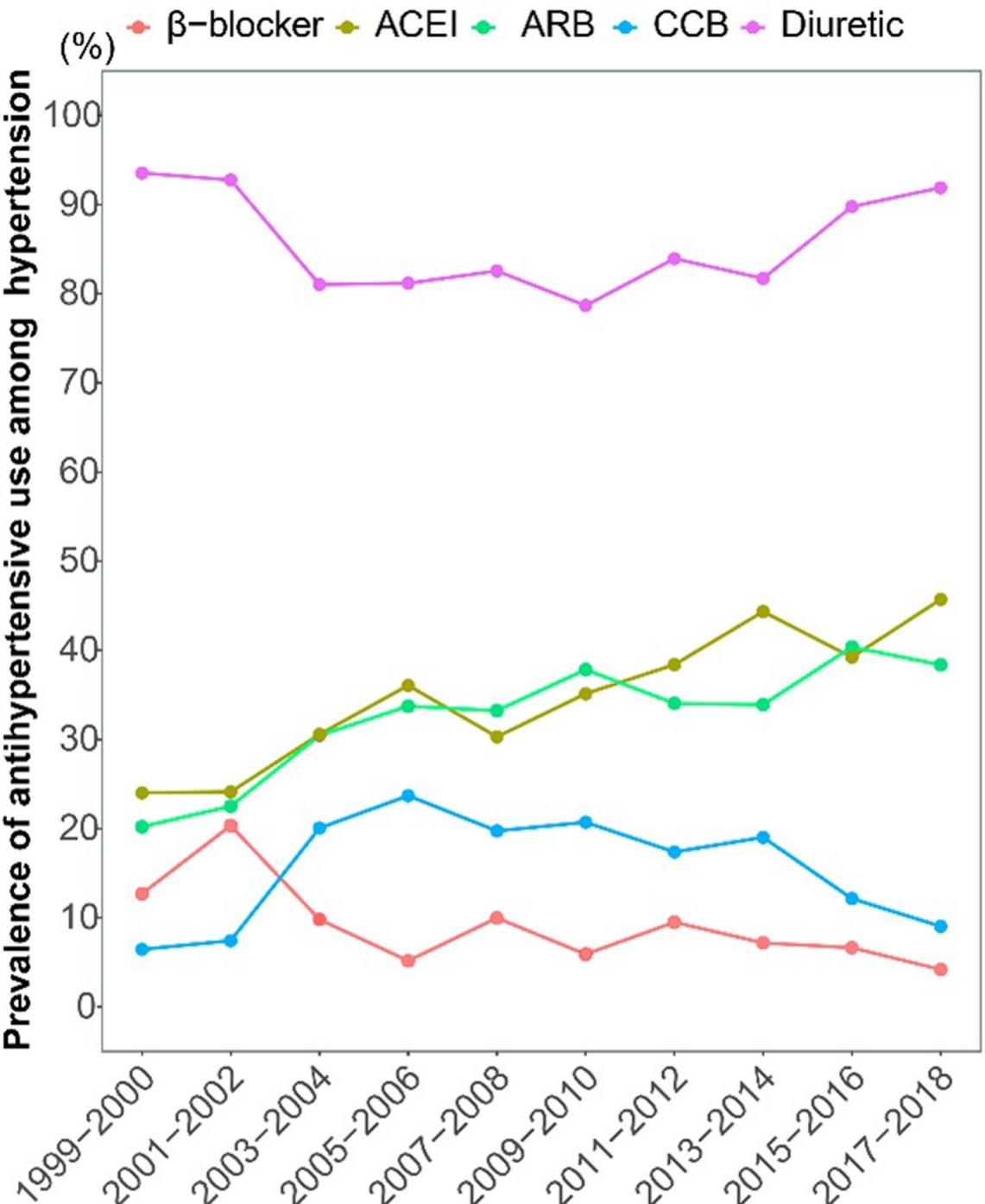

**Fig 3. Prevalence of antihypertensive use among US hypertension adults from 1999 to 2018.** ACEI: angiotensin-converting enzyme inhibitors; ARB: angiotensin receptor blockers; CCB: calcium channel blocker.

Our results showed that from the age distribution, the prevalence of hypertension in the group of ≥40 years old presented an increasing trend from 1999 to 2018, and the prevalence of hypertension in adults in each survey year increased with the increase of age. Many studies have found that aging is a risk factor for hypertension, indicating that the elderly is still the key

**Table 3. Influencing factors of awareness, treatment, and control among US hypertension adults from 1999 to 2018.**

| Characteristics | | Awareness | | Treatment | | Control | | Un-control of blood pressure in persons with antihypertensive drugs | |
|---|---|---|---|---|---|---|---|---|---|
| | | OR (95% CI) | P | OR (95% CI) | P | OR (95% CI) | P | OR (95% CI) | P |
| **Age (years)** | 20–49 | 1 (reference) | | 1 (reference) | | 1 (reference) | | 1 (reference) | |
| | 40–59 | 1.388 (1.197,1.608) | <**0.0001** | 4.249 (3.692,4.891) | <**0.0001** | 0.907 (0.802,1.027) | **0.122** | 1.091 (0.849,1.403) | 0.493 |
| | 60–85 | 1.415 (1.216,1.647) | <**0.0001** | 8.661 (7.518,9.978) | <**0.0001** | 0.655 (0.579,0.740) | <**0.0001** | 1.915 (1.495,2.454) | <**0.0001** |
| **Sex** | Female | 1 (reference) | | 1 (reference) | | 1 (reference) | | 1 (reference) | |
| | Male | 0.789 (0.704,0.883) | <**0.0001** | 0.767 (0.707,0.833) | <**0.0001** | 0.891 (0.822,0.966) | **0.006** | 0.914 (0.829,1.008) | 0.072 |
| **Race** | Non-Hispanic white | 1 (reference) | | 1 (reference) | | 1 (reference) | | 1 (reference) | |
| | Non-Hispanic black | 1.220 (1.063,1.400) | **0.005** | 1.537 (1.378,1.714) | <**0.0001** | 0.810 (0.736,0.891) | <**0.0001** | 1.436 (1.299,1.587) | <**0.0001** |
| | Mexican American | 0.753 (0.653,0.868) | <**0.001** | 0.832 (0.725,0.955) | **0.009** | 0.846 (0.738,0.971) | **0.017** | 1.150 (0.985,1.342) | 0.076 |
| | Other Hispanic | 0.926 (0.699,1.227) | 0.590 | 0.836 (0.680,1.027) | 0.088 | 0.881 (0.723,1.073) | 0.207 | 1.214 (0.984,1.497) | 0.070 |
| | Other race | 0.895 (0.737,1.086) | 0.258 | 0.921 (0.777,1.092) | 0.343 | 0.821 (0.706,0.954) | **0.010** | 1.245 (1.018,1.524) | **0.033** |
| **Marital status** | No | 1 (reference) | | 1 (reference) | | 1 (reference) | | 1 (reference) | |
| | Yes | 1.064 (0.958,1.181) | 0.243 | 1.192 (1.086,1.309) | <**0.001** | 1.095 (1.008,1.189) | **0.032** | 0.925 (0.817,1.047) | 0.217 |
| **Education level** | Less than high school | 1 (reference) | | 1 (reference) | | 1 (reference) | | 1 (reference) | |
| | High school graduate | 0.990 (0.864,1.134) | 0.881 | 0.982 (0.868,1.111) | 0.770 | 1.023 (0.926,1.130) | 0.653 | 0.984 (0.862,1.124) | 0.815 |
| | Some college | 1.139 (0.983,1.320) | 0.083 | 1.002 (0.878,1.143) | 0.977 | 1.059 (0.946,1.186) | 0.316 | 1.008 (0.877,1.159) | 0.908 |
| | College graduate or above | 0.906 (0.765,1.073) | 0.250 | 0.885 (0.758,1.032) | 0.119 | 1.013 (0.896,1.145) | 0.837 | 0.914 (0.777,1.075) | 0.276 |
| **Health insurance** | No | 1 (reference) | | 1 (reference) | | 1 (reference) | | 1 (reference) | |
| | Yes | 1.428 (1.213,1.680) | <**0.0001** | 2.313 (2.019,2.649) | <**0.0001** | 1.364 (1.195,1.557) | <**0.0001** | 0.769 (0.627,0.945) | **0.013** |
| **PIR** | Low income | 1 (reference) | | 1 (reference) | | 1 (reference) | | 1 (reference) | |
| | Middle income | 0.914 (0.781,1.070) | 0.261 | 1.030 (0.900,1.178) | 0.669 | 0.895 (0.799,1.002) | 0.054 | 0.981 (0.829,1.161) | 0.823 |
| | High income | 0.788 (0.674,0.921) | **0.003** | 0.946 (0.830,1.077) | 0.397 | 0.891 (0.787,1.008) | 0.067 | 0.921 (0.782,1.084) | 0.318 |
| **Life's simple 7** | Inadequate health | 1 (reference) | | 1 (reference) | | 1 (reference) | | 1 (reference) | |
| | Average health | 1.111 (0.981,1.258) | 0.096 | 0.943 (0.842,1.055) | 0.303 | 2.035 (1.834,2.258) | <**0.0001** | 0.479 (0.423,0.542) | <**0.0001** |
| | Optimal health | 2.100 (1.623,2.717) | <**0.0001** | 0.834 (0.666,1.044) | 0.112 | 6.669 (5.522,8.055) | <**0.0001** | 0.124 (0.091,0.169) | <**0.0001** |

PIR: income-to-poverty ratio

monitoring group for hypertension [25]. From the gender distribution, the crude prevalence of hypertension in male adults showed an increasing trend from 1999 to 2018. While, there was no statistical significance in the variation trend of female. In addition, the gender of male has been confirmed as an influencing factor of hypertension among US adults in this study. This may be related to unhealthy eating habits, high mental pressure and reduced physical

labor of men [26]. However, women are more likely to accept the concept of healthy life and perform better than men in hypertension prevention behaviors (such as blood pressure monitoring and weight control, etc.), resulting in a higher prevalence of hypertension in men [27]. Therefore, health education related to hypertension should be strengthened among men. In addition, non-Hispanic black was the shared factors that increased the rate of incidence, awareness, treatment, and un-control for hypertension. The relationship between race and hypertension has received increasing attention [28]. There have been many reports on the relationship between race and hypertension, but the relationship between race and hypertension has not been fully clarified. Genetics and lifestyle differences are probably the most important factors [29].

In our study, we also found that lower income and education were positively associated with the occurrence of hypertension. In general, low-income hypertensive patients have relatively low purchasing power for fruits and vegetables and other food, unreasonable diet structure, and relatively little time and facilities for physical exercise, so they have poor ability to control blood pressure [30, 31]. Secondly, low-income patients are generally not well educated, often lack knowledge of health care and hypertension-related diseases, and lack of attention to their own health, unable to do scientific and regular exercise and diet, and cannot detect disease risk factors, so hypertension progresses quickly and the condition is serious [32]. Interestingly, we found that health insurance was an influencing factor of awareness, treatment, and control among US hypertension adults. This may be owing to people with insurance tend to have higher incomes and greater financial ability to value physical health [33]. A healthy lifestyle is also essential for blood pressure control. Generally, a healthy lifestyle includes eating right, exercising moderately, quitting smoking, abstaining from alcohol, and maintaining mental balance [3]. In this study, optimal health has been confirmed to be an advocated approach to improve the burden of hypertension.

Our study had several strengths. First, the complex sampling design of NHANES and survey weighted analysis of our study permitted the accurately calculation of prevalence estimates for the US population. The large sample size of NHANES facilitated subgroup analyses for the burden of hypertension. Second, we updated the tendency in the prevalence, awareness, treatment, and control of hypertension nationally represented US adults from 1999 to 2018, as well as in the subgroup of social demography and LS7. Third, based on the epidemiological characteristics of hypertension, we explored the influencing factors of its prevalence, awareness, treatment, and control to provide theoretical basis for the health policy.

This study has several limitations. Firstly, there are non-response bias and recall bias during the whole survey period in NHANES. For non-responders, we don't know the prevalence and management of hypertension. Meanwhile, the inaccurately reporting that they did not remember whether their doctor had told them they had hypertension and misreported the status of anti-hypertensive medication, may lead biased estimation. Secondly, for the analysis of influencing factors, residual confounding and reverse causation may existed since this is a cross-sectional study. Thirdly, due to a lower number of participants who reported or showed the specific hypertension drugs, the research about the prevalence and influence of antihypertensive strategy was insufficient.

## Conclusion

The rate of newly diagnosed hypertension in US has slightly declined since 2010, but the prevalence of hypertension is on the rise, and the control and treatment of hypertension are still inadequate. To manage hypertension effectively, we need to focus on screening and prevention

for high-risk populations, while advocating for optimal health to improve the burden of hypertension.

## Supporting information

**S1 File. Contents of supplement.**
(DOCX)

## Acknowledgments

The authors expressed gratitude to NHANES for data collection and quality control.

## Author Contributions

**Conceptualization:** Jian Yang.

**Formal analysis:** Chaojun Yang.

**Funding acquisition:** Zhixing Fan, Jian Yang.

**Methodology:** Chaojun Yang.

**Project administration:** Jian Yang.

**Software:** Chaojun Yang.

**Visualization:** Chaojun Yang.

**Writing – original draft:** Zhixing Fan.

**Writing – review & editing:** Zhixing Fan, Jing Zhang, Yifan Huang, Ying Yang, Ping Zeng, Wanyin Cai, Zujin Xiang, Jingyi Wu.

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
