## [Decision Letter · Decision Letter 0]

24 Aug 2023

PONE-D-23-23791Trends and Influence Factors in the Prevalence, Awareness, Treatment, and Control of Hypertension among US Adults from 1999 to 2018PLOS ONE

Dear Dr. Yang,

Thank you for submitting your manuscript to PLOS ONE. After careful consideration, we feel that it has merit but does not fully meet PLOS ONE’s publication criteria as it currently stands. Therefore, we invite you to submit a revised version of the manuscript that addresses the points raised during the review process. Please address comments and revisions requested by reviewers. Please submit your revised manuscript by Oct 08 2023 11:59PM. If you will need more time than this to complete your revisions, please reply to this message or contact the journal office at plosone@plos.org. Please include the following items when submitting your revised manuscript:A rebuttal letter that responds to each point raised by the academic editor and reviewer(s). You should upload this letter as a separate file labeled 'Response to Reviewers'.A marked-up copy of your manuscript that highlights changes made to the original version. You should upload this as a separate file labeled 'Revised Manuscript with Track Changes'.An unmarked version of your revised paper without tracked changes. You should upload this as a separate file labeled 'Manuscript'.If applicable, we recommend that you deposit your laboratory protocols in protocols.io to enhance the reproducibility of your results. Protocols.io assigns your protocol its own identifier (DOI) so that it can be cited independently in the future. For instructions see: https://journals.plos.org/plosone/s/submission-guidelines#loc-laboratory-protocols. Additionally, PLOS ONE offers an option for publishing peer-reviewed Lab Protocol articles, which describe protocols hosted on protocols.io. Read more information on sharing protocols at https://plos.org/protocols?utm_medium=editorial-email&utm_source=authorletters&utm_campaign=protocols.

We look forward to receiving your revised manuscript.

Kind regards,

Hean Teik Ong

Academic Editor

PLOS ONE

Journal Requirements:

Additional Editor Comments:

Please address comments and revisions requested by reviewers.

Reviewers' comments:

Reviewer's Responses to Questions

**Comments to the Author**

1. Is the manuscript technically sound, and do the data support the conclusions?

Reviewer #1: Yes

Reviewer #2: Yes

2. Has the statistical analysis been performed appropriately and rigorously? 

Reviewer #1: I Don't Know

Reviewer #2: Yes

3. Have the authors made all data underlying the findings in their manuscript fully available?

Reviewer #1: Yes

Reviewer #2: Yes

4. Is the manuscript presented in an intelligible fashion and written in standard English?

Reviewer #1: Yes

Reviewer #2: Yes

5. Review Comments to the Author

Reviewer #1: Regarding this statement in page 4

However, there is no estimation of the trends in the prevalence, awareness, treatment, and control of hypertension over a 20-years period in the US, and few studies have explored the factors influencing hypertension incidence, awareness, treatment, and control simultaneously.

There are 2 other articles that cover the same population over an almost similar period of time

1) Trends in the Prevalence, Awareness, Treatment, and Control of Hypertension among Young Adults in the United States, 1999–2014

Yiyi Zhang, and Andrew E Moran, MD. Hypertension. 2017 Oct; 70(4): 736–742.

2)Trends in Blood Pressure Control Among US Adults With Hypertension, 1999-2000 to 2017-2018

Paul Muntner, Shakia T Hardy, Lawrence J Fine , Byron C Jaeger, Gregory Wozniak, Emily B Levitan , Lisandro D Colantonio. JAMA 2020 Sep 22;324(12)

Suggest to review that statement.

Reviewer #2: This is a well researched article that needs minor revision to make it more useful and easier reading for readers.

1. All abbreviations in the Abstract, Text and Legends should be preceded by full spelling. To make it easier for readers, a list of abbreviations should be added before the Introduction of the main text.

2. In calculating the average annual percent change (AAPC), the author assumes that the trend is consistent over the period studied, with a statistical calculation used to give the average rate. However as can be seen from Figure 1, there is no consistent trend over the period studied. For example, for newly diagnosed hypertension, the incidence dropped from 1999-2014, and then rose from 2014-2018. Authors therefore need to rewrite the section on "Trends in the prevalence of awareness, treatment, and control among hypertension", to apply AAPC selectively over specific periods or not to use it for certain parameters. It is very important not to over, under or incorrectly estimate consistencies in the trend.

3. Figure 1 is most important, and graphically represents what the whole article is about. It should be presented as 3 separate Figures.

6. PLOS authors have the option to publish the peer review history of their article (what does this mean?). If published, this will include your full peer review and any attached files.

Reviewer #1: No

Reviewer #2: **Yes: **Hean Teik Ong

---

## [Author Response · Author response to Decision Letter 0]

30 Aug 2023

Reviewer #1: 

Regarding this statement in page 4

However, there is no estimation of the trends in the prevalence, awareness, treatment, and control of hypertension over a 20-years period in the US, and few studies have explored the factors influencing hypertension incidence, awareness, treatment, and control simultaneously.

There are 2 other articles that cover the same population over an almost similar period of time

1) Trends in the Prevalence, Awareness, Treatment, and Control of Hypertension among Young Adults in the United States, 1999–2014

Yiyi Zhang, and Andrew E Moran, MD. Hypertension. 2017 Oct; 70(4): 736–742.

2)Trends in Blood Pressure Control Among US Adults With Hypertension, 1999-2000 to 2017-2018

Paul Muntner, Shakia T Hardy, Lawrence J Fine , Byron C Jaeger, Gregory Wozniak, Emily B Levitan , Lisandro D Colantonio. JAMA 2020 Sep 22;324(12)

Suggest to review that statement.

Response : We have revised this statement and cited the two reference in the introduction section.

Reviewer #2: 

This is a well researched article that needs minor revision to make it more useful and easier reading for readers.

1. All abbreviations in the Abstract, Text and Legends should be preceded by full spelling. To make it easier for readers, a list of abbreviations should be added before the Introduction of the main text.

Response 1: We have added full spelling about the abbreviations in the Abstract, Text and Legends. A list of abbreviations was added before the Introduction of the main text.

2. In calculating the average annual percent change (AAPC), the author assumes that the trend is consistent over the period studied, with a statistical calculation used to give the average rate. However as can be seen from Figure 1, there is no consistent trend over the period studied. For example, for newly diagnosed hypertension, the incidence dropped from 1999-2014, and then rose from 2014-2018. Authors therefore need to rewrite the section on "Trends in the prevalence of awareness, treatment, and control among hypertension", to apply AAPC selectively over specific periods or not to use it for certain parameters. It is very important not to over, under or incorrectly estimate consistencies in the trend.

Response 2: We used annual percent change (APC) to evaluate the internal trend of each independent interval before and after inflection (Line 166-178), and rewrite the section on "Trends in the prevalence of awareness, treatment, and control among hypertension"(Line 208-211, 233-239, 252-259).

3. Figure 1 is most important, and graphically represents what the whole article is about. It should be presented as 3 separate Figures.

Response 3: We have separated into 3 Figures.

---

## [Decision Letter · Decision Letter 1]

14 Sep 2023

Trends and Influence Factors in the Prevalence, Awareness, Treatment, and Control of Hypertension among US Adults from 1999 to 2018

PONE-D-23-23791R1

Dear Dr. Chaojun Yang,

We’re pleased to inform you that your manuscript has been judged scientifically suitable for publication and will be formally accepted for publication once it meets all outstanding technical requirements.

Kind regards,

Hean Teik Ong

Academic Editor

PLOS ONE

Additional Editor Comments (optional):

Reviewers' comments:

Reviewer's Responses to Questions

**Comments to the Author**

1. If the authors have adequately addressed your comments raised in a previous round of review and you feel that this manuscript is now acceptable for publication, you may indicate that here to bypass the “Comments to the Author” section, enter your conflict of interest statement in the “Confidential to Editor” section, and submit your "Accept" recommendation.

Reviewer #1: All comments have been addressed

Reviewer #2: All comments have been addressed

2. Is the manuscript technically sound, and do the data support the conclusions?

Reviewer #1: Yes

Reviewer #2: Yes

3. Has the statistical analysis been performed appropriately and rigorously? 

Reviewer #1: I Don't Know

Reviewer #2: Yes

4. Have the authors made all data underlying the findings in their manuscript fully available?

Reviewer #1: Yes

Reviewer #2: Yes

5. Is the manuscript presented in an intelligible fashion and written in standard English?

Reviewer #1: Yes

Reviewer #2: Yes

6. Review Comments to the Author

Reviewer #1: This is a well written article. The trends and factors influencing prevalence, awareness, treatment

and control of hypertension were well analysed.

Reviewer #2: This is a revised manuscript, that is well written, with initial comments posing 2 questions asking for minor revisions. The reviewer comments have been adequately addressed and necessary revisions have been done. The article can be accepted.

7. PLOS authors have the option to publish the peer review history of their article (what does this mean?). If published, this will include your full peer review and any attached files.

Reviewer #1: **Yes: **Liew Yew Fong

Reviewer #2: **Yes: **Hean Teik Ong

---

## [Editor Report · Acceptance letter]

19 Sep 2023

PONE-D-23-23791R1 

Trends and Influence Factors in the Prevalence, Awareness, Treatment, and Control of Hypertension among US Adults from 1999 to 2018 

Dear Dr. Yang:

I'm pleased to inform you that your manuscript has been deemed suitable for publication in PLOS ONE. Congratulations! Your manuscript is now with our production department. 

Kind regards, 

on behalf of

Dr. Hean Teik Ong 

Academic Editor

PLOS ONE